METHODS AND RESOURCES

# Meiotic Cas9 expression mediates gene conversion in the male and female mouse germline

**Alexander J. Weitzel**[1], **Hannah A. Grunwald**[1], **Ceri Weber**[1], **Rimma Levina**[1], **Valentino M. Gantz**[1], **Stephen M. Hedrick**[2], **Ethan Bier**[1,3], **Kimberly L. Cooper**[1,3]*

**1** Division of Biological Sciences, Section of Cell and Developmental Biology, University of California San Diego, La Jolla, California, United States of America, **2** Division of Biological Sciences, Section of Molecular Biology, University of California San Diego, La Jolla, California, United States of America, **3** Tata Institute for Genetics and Society, University of California San Diego, La Jolla, California, United States of America

* kcooper@ucsd.edu

**Data Availability Statement:** All primary Sanger sequence traces that confirm the Spo11Cas9-P2A-eGFP knock-in allele and all genotyping data for offspring of the complementation test crosses are

## Abstract

Highly efficient gene conversion systems have the potential to facilitate the study of complex genetic traits using laboratory mice and, if implemented as a "gene drive," to limit loss of biodiversity and disease transmission caused by wild rodent populations. We previously showed that such a system of gene conversion from heterozygous to homozygous after a sequence targeted CRISPR/Cas9 double-strand DNA break (DSB) is feasible in the female mouse germline. In the male germline, however, all DSBs were instead repaired by end joining (EJ) mechanisms to form an "insertion/deletion" (indel) mutation. These observations suggested that timing Cas9 expression to coincide with meiosis I is critical to favor conditions when homologous chromosomes are aligned and interchromosomal homology-directed repair (HDR) mechanisms predominate. Here, using a Cas9 knock-in allele at the *Spo11* locus, we show that meiotic expression of Cas9 does indeed mediate gene conversion in the male as well as in the female germline. However, the low frequency of both HDR and indel mutation in both male and female germlines suggests that Cas9 may be expressed from the *Spo11* locus at levels too low for efficient DSB formation. We suggest that more robust Cas9 expression initiated during early meiosis I may improve the efficiency of gene conversion and further increase the rate of "super-mendelian" inheritance from both male and female mice.

## Introduction

The mouse remains the single most utilized laboratory model of human physiology and disease. Yet, breeding strategies that combine multiple homozygous transgenic or knock-in alleles are cumbersome in mice due to small litter sizes and long generation times relative to other traditional model systems. Practical challenges therefore limit the full potential for mouse genetic research to model certain complex human genetic disorders and to engineer models that better mimic human metabolism and immunity for therapeutic drug design. We

available at Zenodo with the identifier 10.5281/
zenodo.5510697. Annotated sequence data for the
Spo11Cas9-P2A-eGFP knock-in allele is available in
GenBank with the accession number OK067273.

**Funding:** This work was funded by a Packard
Fellowship in Science and Engineering, award
number 2015-63114, from the David and Lucile
Packard Foundation and NIH grant R21GM129448
awarded to KLC. The work was also supported by
NIH grant R01AI131081 awarded to SMH and an
Allen Frontiers Group Distinguished Investigators
Award to EB. Work in the laboratories of KLC and
EB was also supported by a gift from the Tata
Trusts in India to TIGS-UCSD and TIGS-India. AJW
and HAG were supported by NIH training grant
T32GM007240, and RL was supported by NIH
training grant T32GM133351. The funders had no
role in study design, data collection and analysis,
decision to publish, or preparation of the
manuscript.

**Competing interests:** I have read the journal's
policy and the authors of this manuscript have the
following competing interests: KLC, VMG, SMH,
and EB serve as science advisory board members
for Synbal, Inc. All other authors declare they have
no competing interests.

**Abbreviations:** DSB, double-strand DNA break;
E16.5, embryonic day 16.5; EJ, end joining; gRNA,
guide RNA; HDR, homology-directed repair; IF,
immunofluorescence; MMEJ, microhomology-
mediated end joining; NHEJ, nonhomologous end
joining; OCT, optimal cutting temperature.

previously demonstrated that super-mendelian inheritance mediated by CRISPR/Cas9 is feasible through the female germline of mice, increasing the transmission frequency of a transgenic allele [1]. Similar systems have been proposed for implementation as a "gene drive" over multiple generations in wild rodents to limit the spread of infectious disease or to reduce populations of invasive mice and rats in sensitive island ecosystems [2–6].

To assess the feasibility of CRISPR/Cas9–mediated gene conversion, we previously engineered a "CopyCat" transgene that disrupts the fourth exon of *Tyrosinase*, a gene that is required for melanization in mice [7]. The $Tyr^{CopyCat}$ transgene expresses the single guide RNA (gRNA) used to target the site of its own insertion and an mCherry fluorescent marker to track its inheritance. We then used available genetic tools to express Cas9 in the early embryo and in the male and female germline.

There are two "nonconservative" repair outcomes of CRISPR/Cas9–mediated double-strand DNA breaks (DSBs) that alter the wild-type target sequence in a $Tyr^{CopyCat}$ heterozygous animal. First, the two ends may be rejoined by nonhomologous end joining or microhomology mediated end joining [NHEJ or MMEJ, hereafter referred to as "end joining" (EJ)], which can cause insertion or deletion mutations (indels) that disrupt the gRNA target site and prevent subsequent cutting (reviewed in [8]). Alternatively, the DSB can be repaired by homology-directed repair (HDR) using genomic sequences that flank the $Tyr^{CopyCat}$ transgene on the homologous chromosome, referred to as "gene conversion" or "homing." If interchromosomal HDR occurs, then the transgene that was inherited on only one chromosome is copied into the DSB of the homologous locus such that its genotype is converted from heterozygosity to homozygosity [1].

CRISPR/Cas9–mediated gene conversion is highly efficient (>90%) in Dipteran insects and in yeast [9–12]. Using multiple Cre/lox strategies to initiate Cas9 expression in oogonia and spermatogonia of mice [13,14], we previously demonstrated gene conversion in the female but not in the male germline [1]. The frequency of $Tyr^{CopyCat}$ allele transmission increased from 50%, predicted by mendelian inheritance, to an average of 72% among offspring of 5 females after CRISPR/Cas9–mediated gene conversion in the germline. We also recombined ultra-tightly linked loci (approximately 9 kb apart) by gene conversion in 22.5% of offspring of females. Although this was an important demonstration of the powerful potential of such approaches, use of two transgenes (Cre and conditional Cas9) to control the timing of Cas9 expression is not an optimal approach to simplify complex crossing schemes.

The observation that gene conversion occurred exclusively in the female germline suggested that sex differences in the progression of gonial development might be key. Oogonia rapidly mature into primary oocytes that initiate meiosis during fetal development [15], while spermatogonia are mitotic throughout the life of the animal and sporadically produce cohorts of meiotic primary spermatocytes [16,17]. In contrast to mitosis, homologous chromosomes are aligned during meiosis I, and the molecular machinery that facilitates interchromosomal HDR predominates over EJ [18]; either or both of these conditions might allow gene conversion to occur in meiotic but not in mitotic cells. We therefore interpreted our data to suggest that timing Cas9 expression to coincide with meiosis I may be critical to allow for gene conversion in both sexes.

Here, we use a single transgene to evaluate the importance of initiating Cas9 expression during prophase of meiosis I for gene conversion to occur in mice. We engineered a knock-in allele at the *Spo11* locus to express Cas9 in conjunction with endogenous SPO11 using a P2A self-cleaving peptide (Fig 1A). Spo11 is the endonuclease and topoisomerase that catalyzes DSB formation required for crossing over during meiotic recombination in prophase of meiosis I [19,20]. For this analysis, the timing but not the catalytic activity of *Spo11* expression is important. In mammals, *Spo11* is expressed only in mature testes and in embryonic ovaries,

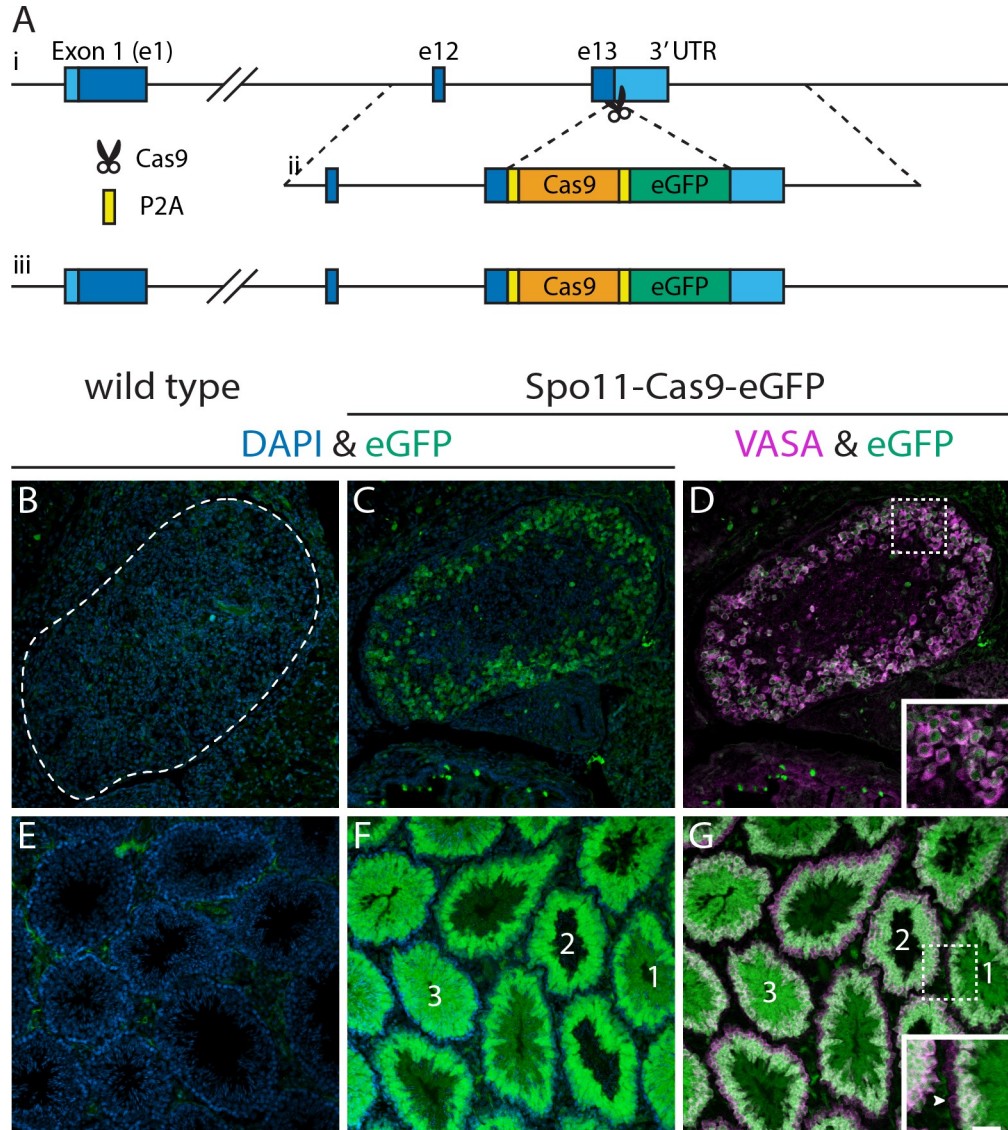

**Fig 1. Male and female germ cell characterization of $Spo11^{Cas9-P2A-eGFP}$ knock-in expression. (A)** Schematic of targeting the $Spo11^{Cas9-P2A-eGFP}$ knock-in allele. (i) Wild-type $Spo11$ locus. (ii) Construct for insertion. (iii) Final $Spo11^{Cas9-P2A-eGFP}$ knock-in locus. The genotyping strategy is depicted in S7 Fig. **(B–G)** Immunofluorescent detection of eGFP (green) and VASA (magenta) plus DAPI (blue) in $Spo11^{+/+}$ (B, E) and $Spo11^{Cas9-P2A-eGFP/+}$ (C,D; F,G) mice. (B–D) embryonic ovaries at E18.5 ($n$ = 3) and (E–G) adult testes ($n$ = 3). Seminiferous tubules labeled 1 -> 3 are in different stages of maturation as determined by chromatin compaction (DAPI) and relative level of VASA expression in outer perimeter cells. The inset shows outer perimeter cells of tubule 1 (arrowhead points to spermatogonium) that express low VASA and no eGFP. By contrast, the outer perimeter cells of tubule 2 (meiotic primary spermatocytes) express higher VASA and low eGFP, and outer perimeter cells of tubule 3 express the highest eGFP. Scale bar: 50 μm for (B–G) and 25 μm for the insets in (D) and (G). eGFP channel brightness was adjusted in (B–D) independent of (E–G); equivalent exposures and all biological replicates are shown in S4 Fig.

and its expression peaks during prophase of meiosis I when homologous chromosomes are aligned [21,22].

Using the same $Tyr^{CopyCat}$ allele and crossing scheme to assess gene conversion efficiency as in our prior work [1], we demonstrate that Cas9 expression driven by $Spo11$ during meiosis I promotes gene conversion in the male as well as female germline. Compared to our most

efficient prior strategy, the relative fraction of detectable DSB repair that is mediated by HDR [(HDR/(HDR + EJ)] is increased in the male (from 0% to 11%) and is similar to what we previously observed in the female (71% versus 67%) consistent with our hypothesis that restricting DSB repair to meiosis I favors HDR. However, total rates of DSB repair (HDR + EJ) are lower in both sexes resulting in absolute gene conversion frequencies that are low in males and reduced in females compared to previous strategies. We suggest that the level of Cas9 expression driven from the *Spo11* locus may be too low for efficient DSB formation, although the timing of expression overlaps the window when interchromosomal HDR does occur. These results demonstrate that gene conversion is possible in the male mouse germline and emphasize the importance of both timing and levels of Cas9 expression for efficient gene conversion systems in rodents.

## Results

We hypothesized that restricting Cas9 expression to meiosis I in both the male and female germline should result in gene conversion events in both sexes. *Spo11* expression has been detected in male germ cells isolated by meiotic stage using FACS and in a precisely staged series of meiotic embryonic ovaries [21]. Expression is initiated in leptotene primary spermatocytes and oocytes and increases during zygotene [21,22]. In males, *Spo11* expression continues to rise through pachytene and diplotene and can be detected in round spermatids but not in residual bodies of elongating spermatids [22]. In females, expression falls through pachytene and diplotene, when oocytes arrest until ovulation [21].

We used a knock-in approach to place Cas9 after the final exon of *Spo11* to take advantage of the endogenous timing of expression from this locus while attempting to minimize disruption of SPO11, which is critical for fertility in mice [23]. Hereafter, we refer to this allele as *Spo11*$^{Cas9-P2A-eGFP}$. PCR amplification and Sanger sequencing confirm the knock-in construct was correctly targeted to the *Spo11* locus (S1 Fig). SPO11, Cas9, and eGFP (to visually report expression from the locus) are each separated by self-cleaving peptide sequences (Fig 1A), and each protein is detectable by western blot as a single band of appropriate size unique to heterozygous animals (S2 Fig). Although we did not detect a significant difference in the normalized level of SPO11 expression in *Spo11*$^{Cas9-P2A-eGFP/+}$ adult testes compared to wild type (S2A' Fig), we recovered no offspring from a homozygous male after 20 months of continuous harem mating. Section histology of the testes revealed this homozygous *Spo11*$^{Cas9-P2A-eGFP}$ male mouse had small testes and defects of the seminiferous tubule (S3 Fig) similar to *Spo11*$^{-/-}$ mice [23]. All further experiments were performed using heterozygous animals, which were viable and fertile, consistent with *Spo11* haplosufficiency.

We predicted that the regulatory context of the *Spo11* locus would limit Cas9-P2A-eGFP expression to meiotic spermatocytes and oocytes. We tested this hypothesis by performing immunofluorescence (IF) to detect eGFP expression in tissue sections of adult testes and embryonic ovaries. In males, waves of spermatogenesis pulse along the length of seminiferous tubules such that a single cross section captures multiple phases [24,25]. Meiotic and maturing spermatocytes strongly express eGFP, which was not detected in cohorts of mitotically expanding spermatogonia identified in some tubules as an outer ring of cells with loosely compacted chromatin (DAPI) and low VASA expression [26,27] (Fig 1E–1G). In females, eGFP expression is first detected in ovaries at embryonic day 16.5 (E16.5), coinciding with early zygotene [21]. Expression increases in ovaries at E17.5 (late zygotene) and persists at E18.5. We note that the level of expression from the Spo11-Cas9-eGFP transgene appears to be higher in the male germline than in the female germline as determined by intensity of eGFP detection by IF (Fig 1B–1G, S4 Fig).

To assess the efficiency of gene conversion using this transgene, we first crossed heterozygous $Spo11^{Cas9-P2A-eGFP}$ knock-in mice with homozygous $Tyrosinase^{chinchilla}$ ($Tyr^{ch/ch}$) mice to introduce an ultra-tightly linked SNP in exon 5 of *Tyrosinase*. The $Tyr^{ch}$ SNP is easily identified by PCR followed by Sanger sequencing and marks the recipient chromosome that will be targeted for DSB formation, allowing us to detect even a single gene conversion event. We backcrossed some of these animals to $Tyr^{ch/ch}$ to homozygose the marked recipient chromosome. We then crossed $Spo11^{Cas9-P2A-eGFP/+};Tyr^{ch/ch}$ (or $Tyr^{ch/+}$) male mice to $Tyr^{CopyCat/+}$ female mice (Fig 2B) to limit the possibility of maternal Cas9 transmission that might induce indel mutations in the early zygote [1].

Cas9 expressed from this transgene in meiotic cells complexes with the Tyr4a-gRNA encoded by the $Tyr^{CopyCat}$ transgene, which targets exon 4 of *Tyr* only on the $Tyr^{ch}$-marked recipient chromosome (Fig 2A). A DSB may then be nonconservatively repaired by EJ to form an indel or by interchromosomal HDR to result in gene conversion. If gene conversion occurs, the $Tyr^{CopyCat}$ transgene will be encoded in *cis* with the $Tyr^{ch}$ SNP allele, which is expected to rarely occur between these loci by natural recombination with a probability of $4.7 \times 10^{-5}$.

To determine the frequency of these outcomes by genetic complementation, we crossed $Spo11^{Cas9-P2A-eGFP/+}; Tyr^{CopyCat/ch}$ mice to $Tyr^{null/null}$ albino mice that have a null mutation in *Tyrosinase* exon 1 (Fig 2C). We genotyped all offspring of this cross for the $Tyr^{ch}$ SNP (Fig 2D) and assessed coat color and mCherry fluorescence (S5 Fig) of the $Tyr^{ch}$-positive population. Gray mice represent no cut or an in-frame repair at the gRNA target site in exon 4 that preserves activity of the $Tyr^{ch}$ allele, while white mice inherited a null mutation on the recipient $Tyr^{ch}$ chromosome that fails to complement the $Tyr^{null}$ allele. Of these, we identified white mice that also fluoresce red due to inheritance of the $Tyr^{CopyCat}$ transgene that disrupts *Tyr* function (Fig 2D, right-most mouse). Genotyping these red fluorescent $Tyr^{ch}$ mice confirmed inheritance of the $Tyr^{CopyCat}$ allele (S6 Fig). Since the $Tyr^{null}$ mouse could contribute neither the $Tyr^{ch}$ SNP nor the $Tyr^{CopyCat}$ transgene, a mouse with both alleles must have inherited them on a single parental chromosome resulting from a germline interchromosomal HDR event.

We assessed the progeny of 5 males and 5 females using this complementation test cross strategy (Fig 2C and 2D). In total, we identified 263 $Tyr^{ch}$-positive offspring of males and 73 $Tyr^{ch}$-positive offspring of females. Of these, 8 offspring of males (3% of $Tyr^{ch}$) and 4 offspring of females (6% of $Tyr^{ch}$) inherited the $Tyr^{CopyCat}$ transgene by gene conversion. Although these numbers are substantially lower than the average 44% gene conversion we previously reported in the female germline (*Vasa-Cre;H11-LSL-Cas9* strategy), this is the first reported observation of CRISPR/Cas9–mediated gene conversion in the male germline.

Notably, Cas9 driven from the *Spo11* locus reduced the absolute rate of nonconservative DSB repair [(HDR + EJ)/total $Tyr^{ch}$] compared to our previous *Vasa-Cre;H11-LSL-Cas9* conditional strategy from 100% to 29% in the male germline (Fig 3A) and from 60% to 8% in the female germline (Fig 3B). This reduction in DSB formation suggests the overall CRISPR-based cleavage efficiency is lower using the $Spo11^{Cas9-P2A-eGFP}$ transgene than for previously employed strategies based on *Vasa*-mediated conditional Cas9 expression. We note, however, that the proportion of HDR events among all DSB repair outcomes [HDR/(HDR + EJ)] remains similar in females (71% in the previous study versus 67% in this study) and disproportionately improves in males (from 0% to 11%).

## Discussion

Although moderately efficient gene conversion in one parent might be sufficient to increase the probability of obtaining complex multilocus genotypes, such approaches would be vastly

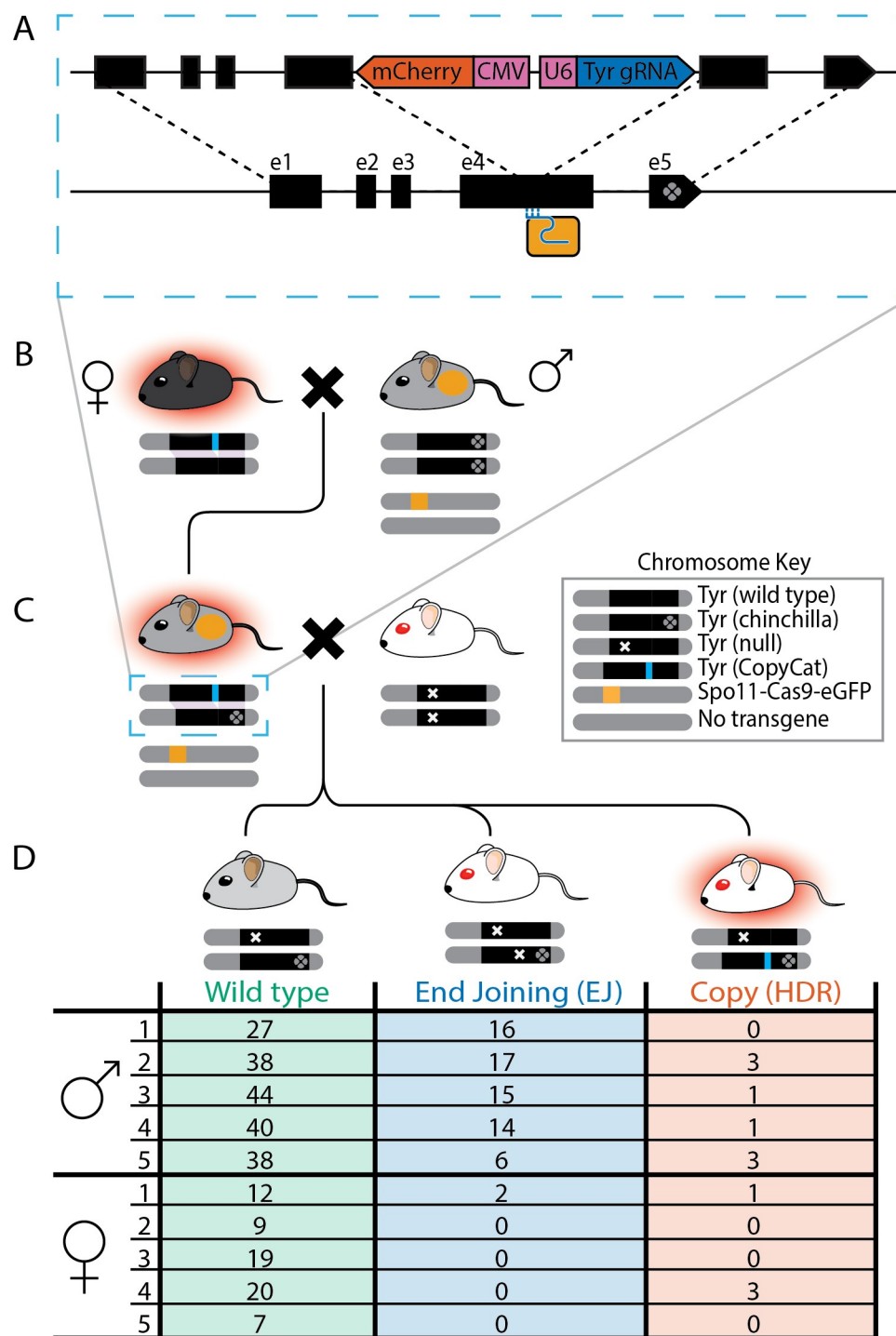

**Fig 2. Breeding scheme to achieve gene conversion with summary table of results from $Tyr^{ch}$ offspring of complementation test crosses. (A)** Schematic of the genetically encoded $Tyr^{CopyCat}$ transgene in Exon 4 of *Tyrosinase* (top chromosome) and its insertion into the recipient chromosome (bottom) during interchromosomal HDR. mCherry (red), CMV and U6 promoters (pink), and Tyr4a-gRNA (dark blue). Dotted lines indicate *Tyr* homology between chromosomes. Cas9 (light orange box) and gRNA (light blue) only target the wild-type *Tyr* allele. **(B)** Female $Tyr^{CopyCat/+}$ crossed with male $Spo11^{Cas9-P2A-eGFP/+};Tyr^{ch/ch}$ mice. **(C)** Complementation test cross of $Tyr^{CopyCat/ch}$; $Spo11^{Cas9-P2A-eGFP/+}$ male or female mice to albino $Tyr^{null/null}$ mice. **(D)** Numbers of $Tyr^{ch}$ offspring with each genotype in families (1 to 5) from male or female $Tyr^{CopyCat/ch};Spo11^{Cas9-P2A-eGFP/+}$ parents. "Chromosome Key" shows genotypes of pictographic chromosomes under each mouse. The underlying data for each family can be found in the associated Zenodo data repository at (https://doi.org/10.5281/zenodo.5510697). EJ, end joining; gRNA, guide RNA; HDR, homology-directed repair.

improved by gene conversion in both females and in males using a single Cas9 transgene [28]. A previous study employing direct expression from a single *Vasa-Cas9* transgene did not facilitate detectable gene conversion, although the recipient chromosome was not marked [29]. Here, using a $Spo11^{Cas9-P2A-eGFP}$ transgene, we have demonstrated the first reported gene conversion in the male mouse germline. We also show that gene conversion as a proportion of all repair events in the female germline is similar to our prior strategy. However, total repair events are substantially reduced in both sexes.

Together with our previous findings, our results suggest that the overall gene conversion efficiency is dependent upon two key factors: the absolute rate of nonconservative DSB repair [(HDR + EJ)/total $Tyr^{ch}$] and the relative rate of repair that is HDR [HDR/(HDR + EJ)] shown in Fig 3. Here, we synthesize all of our findings to suggest a model wherein these two rates are influenced by the cumulative amount of Cas9 expression and by the timing of Cas9 expression initiation, respectively. We suggest that both levels and timing of Cas9 dictate the dynamics of Cas9–mediated DSB formation and repair in context of the varied cellular and nuclear environments of male and female mitotic and meiotic germ cells.

The kinetics and mechanisms of DNA break repair have been historically described in the context of ionizing radiation and restriction enzyme-induced DSBs that are frequently repaired to preserve the original allele [30–35]. Recent work has shown that Cas9 induced DSBs may be repaired differently, as the Cas9 protein may remain bound after a DSB is induced and can thus prevent canonical repair machinery from accessing exposed DSBs [36–38], although local chromatin environments can affect Cas9 binding kinetics [39,40]. These mechanics may explain observations that Cas9–induced DSB repair is substantially more error prone than repair after other means of DSB induction [38]. Although DSBs can be infrequently repaired by homologous recombination between sister chromatids after mid to late pachytene of meiosis I [41], the very high frequency of conserved alleles from our crosses suggests that most were never cut as opposed to cut and correctly repaired. Therefore, nonconservatively repaired alleles are a reasonable proxy for total rate of DSB generation.

These studies inform our interpretation of observed differences in the absolute rate of DSB repair in the context of genetically encoded Cas9. In our previous strategies, after Vasa-Cre mediated removal of the lox-STOP-lox, Cas9 was expressed under regulatory control of the strong constitutive CAG promoter at either of the permissive H11 or Rosa26 loci. These strategies generated a nonconservative allele in 100% of $Tyr^{ch}$ male germ cells (Fig 3A, all EJ) and in 60% to 76% of female germ cells [1] (Fig 3B, HDR + EJ). By contrast, the absolute rate of DSB repair in this meiotic $Spo11^{Cas9-P2A-eGFP}$ strategy is substantially lower at 29% (76/263) in male (Fig 3A) and 8% (6/73) in female meiotic germ cells (Fig 3B). Previous studies demonstrated that Cas9–induced DSBs follow the dynamics of mass action; across a fixed window of time, total DSBs increase proportionally to the amount of Cas9 [42], and a higher proportion of DSBs occur earlier in the presence of higher levels of Cas9 [43]. Thus, we hypothesize that the lower absolute rate of DSB repair in the $Spo11^{Cas9-P2A-eGFP}$ strategy may reflect lower Cas9 expression from endogenous *Spo11* regulatory sequences than from the constitutive CAG promoter at a permissive locus. We note that the difference in absolute rate of DSB formation in males and females of the $Spo11^{Cas9-P2A-eGFP}$ strategy correlates with a higher level of eGFP detected in male meiotic germ cells (S4 Fig). This observed difference is also consistent with previously reported quantification of *Spo11* mRNA expression by RT-qPCR, which shows a higher expression level in males than in females [21].

It is essential to note that the total level of Cas9 protein in a cell is dependent not only on the strength of a regulatory sequence, but also on the duration of expression. The 100% absolute rate of DSB repair in the Vasa-Cre;LSL-Cas9 male germline strategies may have resulted from persistent Cas9 expression over several months in the population of self-renewing

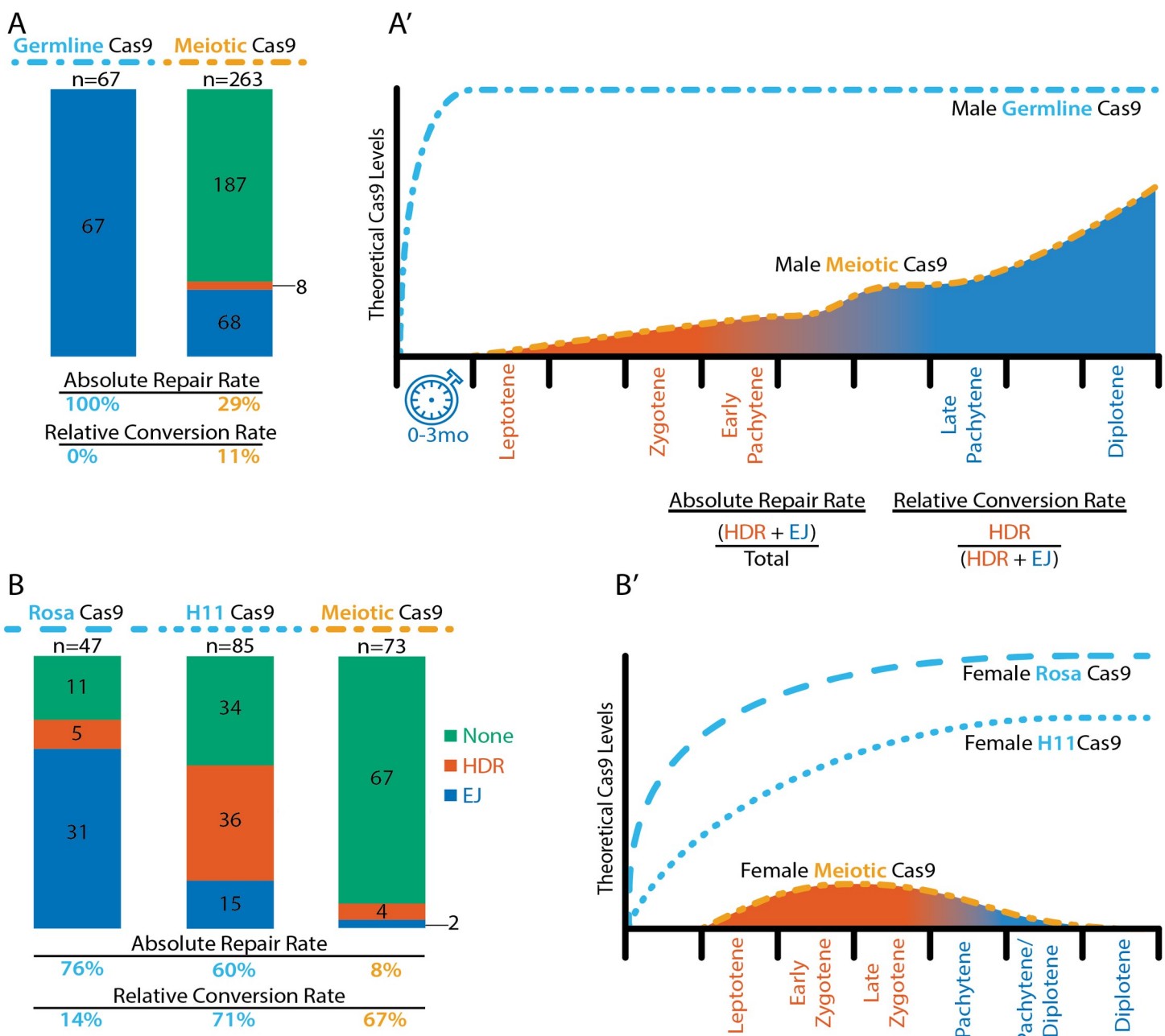

**Fig 3. Comparisons of outcomes of each genetic strategy with models of theoretical Cas9 levels and timing with respect to DSB repair type during meiosis I. (A)** "Germline" Cas9 offspring of males for the combined outcomes of Vasa-Cre;Rosa- and H11-LSLCas9, data sourced from [1], and "Meiotic" Cas9 offspring of males from Fig 2D. **(B)** "Germline" Cas9 offspring of females of each of the Vasa-Cre;Rosa- and H11-LSLCas9 "Germline" strategies and "Meiotic" Cas9 offspring of females from Fig 2D. Each bar graph represents the sum of repair types across all $Tyr^{ch}$ recipient chromosomes in each strategy. Green, no cut or in frame repair; Red, interchromosomal HDR; Blue, EJ. Below each bar graph are the absolute repair rates and relative conversion rates, as defined by the equations shown. **(A' and B')** Models of theoretical Cas9 levels across the range of time relevant to gene conversion. Each hash on the x-axis represents approximately 1 day. Leptotene in the female occurs at about E15.5 and initiates sporadically in maturing males. The orange line represents the theorized relative Cas9 level inferred from RT-qPCR levels [21] and our eGFP IF ovaries and testes. The blue dotted line represents the theoretical expression of Cas9 under a strong CMV promoter and initiated in the germline by expression of Vasa-Cre in spermatogonia and oogonia. The blue dotted line plateaus faster in the male than in the female because many months are represented in the interval (clock) before meiosis is initiated in the male germline. The area under the curve and their respective stages are colored red (HDR) or blue (EJ) to demonstrate the inferred preference for the respective repair type at the time of the cut [18]. The relative shaded area of each color (and thus repair type) under each curve is roughly proportional to the observed "Relative Conversion Rate" shown below the respective "Meiosis" bar graphs in (A) and (B). DSB, double-strand DNA break; EJ, end joining; HDR, homology-directed repair.

mitotic spermatogonia (Fig 3A', blue clock). Such a long duration of Cas9 expression would provide ample time for prevalent DSB repair by EJ; resulting indels would be inherited by primary spermatocytes, thus blocking any subsequent DSB formation and gene conversion at the onset of meiosis. By contrast, the $Spo11^{Cas9-P2A-eGFP}$ transgene delays Cas9 expression by months to be initiated first in meiotic primary spermatocytes (Fig 1G and Fig 3A', orange line), and expression termination is controlled by the endogenous $Spo11$ locus. After rising during meiosis I, $Spo11$ levels drop precipitously in maturing spermatids [22].

Together, the above considerations suggest that the absolute rate of DSB formation correlates with a cumulative level of Cas9 expression that is likely lower in the $Spo11^{Cas9-P2A-eGFP}$ strategy than in the Vasa-Cre;LSL-Cas9 strategies and lower in $Spo11^{Cas9-P2A-eGFP}$ females than in males. In contrast to the absolute rate of DSB repair, we hypothesize that the relative rate of HDR [HDR/(HDR + EJ)] is primarily a function of the timing of Cas9 activity rather than the amount of Cas9 protein. In meiotic cells, particularly in early prophase I (leptotene through mid-pachytene), EJ-specific machinery is not detected; homologous recombination becomes the exclusive repair pathway and predominantly occurs between homologous chromosomes [44,45]. Later in prophase I, after mid-pachytene, EJ again becomes the primary DSB repair choice [18]. Second, the single homologous chromosome that is the template for gene conversion is far more likely to be aligned and accessible during meiosis I, and machinery is expressed that limits recombination to occur between homologous chromosomes [18,46]. From these data, we hypothesize that a permissive window for gene conversion by interchromosomal HDR opens in leptotene and closes in mid-pachytene of meiosis I (Fig 3A' and 3B', red shading), whereas DSBs induced before or after this window may be more likely repaired by EJ. Since SPO11 is necessary for meiotic homologous recombination, our knock-in allele that appears to unintentionally reduce its function could cause lower overall rates of HDR, including gene conversion, during this window. However, we note that $Spo11$ is haplosufficient in mice [23].

The similar relative rate of HDR in females of both genetic strategies may reflect a narrow temporal difference between the onset of Cas9 expression in the $Spo11^{Cas9-P2A-eGFP}$ strategy compared to the Vasa-Cre;LSL-Cas9 strategies, since Vasa-Cre is expressed in oogonia that rapidly mature into meiotic oocytes. In the male Vasa-Cre;LSL-Cas9 strategies, by contrast, DSBs may have been formed and repaired by EJ in most or all mitotic spermatogonia prior to opening the permissive window. Using the $Spo11^{Cas9-P2A-eGFP}$ that initiates expression in meiotic primary spermatocytes may bring the onset of Cas9 expression closer to the permissive window. Therefore, temporally shifting Cas9 to the onset of meiosis in both sexes likely resulted in disproportionate improvement of the relative rate of HDR in males versus females.

Our results mark a significant advance in the feasibility of super-mendelian inheritance strategies in rodents by achieving germline gene conversion in both sexes. We propose that adjusting the onset and increasing the level of Cas9 expression can further improve the absolute rate of DSB repair and the proportion of repair that is gene conversion by HDR in both males and females. This may be achieved by identifying promoters that express at a high level and that fall within or immediately prior to the permissive window for HDR during meiosis I and engineering transgenes that cannot disrupt endogenous gene function. An alternative approach might be to use a promoter that drives strong expression with less temporal specificity and to chemically control the timing of Cas9 activity (e.g., with tamoxifen or trimethoprim [47]). Both wild release applications to constitute a "gene drive" and laboratory applications to facilitate breeding complex genotypes will benefit from further increasing the efficiency of gene conversion by improved genetic strategies. However, improvements that require drug delivery would not be practical for implementation in the wild.

Increasing the absolute rate of DSB repair versus the relative rate of HDR will have different effects on improving gene drive versus laboratory applications. Gene drive efficiency can tolerate a lower absolute rate of repair as long as the proportion of repair that is HDR is very high. Otherwise systems must be designed to select against indels that would be resistant to subsequent gene conversion (reviewed in [48]). In a laboratory setting, where the researcher can genotype to choose animals, a higher rate of EJ is more tolerable so long as the rate of HDR is sufficient to obtain genotypes of interest. While showing that CRISPR/Cas9–mediated gene conversion is possible in both male and female mouse germlines, our work reveals nuances that differ from insects and that must be considered for further refinement and implementation in rodents.

## Methods

### Animals

Production of the $Tyr^{CopyCat}$ transgenic mice was previously described in [1]. $Tyr^{chinchilla}$ mice are strain FVB.129P2-Pde6b+ Tyr$^{c\text{-ch/AntJ}}$, stock number 4828, from the Jackson Laboratory (ME, USA). $Tyr^{null}$ mice are strain CD-1, stock number 022, from Charles River Laboratories (MA, USA). The $Spo11^{Cas9\text{-}P2A\text{-}eGFP}$ knock-in mice were produced by Biocytogen (MA, USA) using CRISPR/Cas9 to insert P2A-Cas9-P2A-GFP flanked by 1.5-kb arms of homology to $Spo11$ exon 13 and the 3′ UTR. Mice were housed in accordance with UCSD Institutional Animal Care and Use Committee protocols and fed on a standard breeders diet. All animal research was conducted under UCSD Animal Welfare Assurance number D16-00020 and approved by Institutional Animal Care and Use Committee protocol number S14014. Humane euthanasia was carried out by low-flow exposure to $CO_2$ until loss of consciousness followed by cervical dislocation as a secondary method.

### Sample collection, immunofluorescence, histology, imaging, and western blotting

We collected testes from adult male $Spo11^{Cas9\text{-}P2A\text{-}eGFP/+}$ and wild-type mice. To obtain meiotic ovaries, we crossed $Spo11^{Cas9\text{-}P2A\text{-}eGFP/+}$ males to CD-1 females. Once fertilization was inferred by detection of a copulatory plug, the female was separated from the breeding male. At the desired embryonic stage, we dissected embryos from the uterus, confirmed the embryonic stage by limb morphological stage [49], bisected the embryo in transverse anterior to the ovaries, and collected the tail tip for PCR genotyping to establish sex and presence of the $Spo11^{Cas9\text{-}P2A\text{-}eGFP}$ transgene. Embryos were fixed in 4% PFA overnight. The following day, they were transferred to 20% sucrose solution until saturated. Embryos were embedded in optimal cutting temperature (OCT) blocks and sectioned at 6-μm thickness. Slides were frozen at -80˚C until processing.

Slides were removed from −80˚C and dried. We performed antigen retrieval for 10′ in 5 μg/ml Proteinase K at room temperature. We then fixed in 4%PFA 1xPBS for 5′ and washed 3× for 10′ in 1x PBS. Slides were then blocked (3% BSA, 5% NGS, 0.1% Triton, 0.02% SDS, in 1X PBS) for 1 to 2 hours at room temperature before addition of primary antibodies diluted in block [chicken anti-eGFP, 1:500, Thermo Fisher catalog # A10262; rabbit anti-human Ddx4 (VASA), 1:250, Abcam catalog # AB13840] and incubated at 4˚C for 18 hours. Slides were then washed 3× for 30′ in PBST (1X PBS, 0.1% Triton) prior to incubation in secondary antibodies [Invitrogen (MA, USA), Alexa-488 goat anti-chicken and Life Technologies (CA, USA), 647 goat anti-rabbit, 1:250 each] in block with 300 nM DAPI [Cell Signaling Technology (MA, USA), catalog # 4083S] and incubated at 4˚C for 18 hours. We then washed 3× for 30′ in PBST and mounted under coverslips in Vectashield (Vector Labs, CA, USA).

Images were collected on Leica TCS SP8 multiphoton confocal microscope. Excitation frequency GFP (488 nm): emission range 505 to 550 nm. Excitation frequency Vasa (647 nm): emission range 655 to 685 nm. Excitation frequency DAPI (405 nm): emission range 415 to 470 nm.

The pixel intensity was determined in FIJI by ZProjection "average intensity" for each channel. The brightness of eGFP signal intensity was adjusted equivalently across all samples (Fig 1, S4 Fig) with the exception of the testis in Fig 1F and 1G, which was reduced for clarity.

For hematoxylin and eosin staining of slides, the staining agent was Harris hematoxylin solution. Samples were differentiated in 1% acid alcohol. The bluing agent was saturated lithium carbonate and slides were counterstained in a solution of 0.1% eosin Y and 0.1% phloxine B in acid alcohol. The samples were then dehydrated to 100% EtOH and cleared in xylenes. The slides were mounted under coverslips in Permount and cured for 24 hours. Samples were imaged at 20× magnification under brightfield on an Olympus (Tokyo, Japan), BX61 microscope.

Mouse testis lysate was acquired from dissections of mice at approximately 6 months of age ($n$ = 3 wild type, $n$ = 3 Spo11-Cas9-eGFP heterozygotes) and placed into a lysis solution (8 M Urea, 75 mM NaCl, 50 mM Tris pH 8.0, 1 mM NaF, 1 mM beta-Gly). Each sample was dounce homogenized at 4˚C, sonicated 3× 10 seconds at 4˚C, and centrifuged at 4˚C for 10 minutes at 15,000 g. Protein quantification was achieved via Pierce BCA Protein Assay Kit (Thermo Fisher (MA, USA), #23225).

For western blotting, three 4% to 20% MP TGX Stain-Free Gel (Bio-Rad # 4568093) were loaded with Precision Plus (Bio-Rad (CA, USA), # 1610374), MagicMark (Thermo Fisher #LC5602), and 30 ug of testis lysate. Each gel underwent elecrophoresis for 1.5 hours at 30 mA, 100 V, and transferred to a 0.2 uM nitrocellulose membrane at 60 V with variable ampere at 4˚C for 1.5 hours. Membranes were Ponceau stained to verify transfer and rinsed 2x in ddH2O. For blocking, membranes were washed for 5 minutes at RT with 1X TBS, incubated in block (1X TBS, 0.1% Tween, 5% nonfat dry milk) for 1 hour at RT and washed 3× in TBST (1X TBS, 0.1% Tween) for 5 minutes. For detection using mouse anti-Cas9 in mouse tissues, the block included 20 ug/mL Affinipure Fab Fragment (Jackson ImmunoResearch (PA, USA), catalog # 115-007-003) to reduce background detection of mouse IgG. Primary antibodies were diluted [rabbit anti-Spo11, 1:500, Sigma Aldrich catalog # MABE1167; chicken anti-GFP 1:1,000, Thermo Fisher catalog # A10262; mouse anti-Cas9 1:1,000, Cell Signaling catalog # 14697] in TBST with 5% nonfat dry milk and incubated with membranes for 18 hours at 4˚C followed by washes 3× for 5 minutes in TBST. HRP-conjugated secondary antibodies were diluted [anti-rabbit 1:1,000, Cell Signaling catalog # 7074; anti-chicken 1:500, Thermo Fisher catalog # A16054; anti-mouse 1:1,000, Cell Signaling catalog # 7076] in TBST, and membranes were incubated for 1 hour at RT. Detection was accomplished using the Clarity Max ECL Substrate (Bio-Rad #1705060), and membranes were imaged on a Bio-Rad ChemiDoc XRS+ Gel Imaging System.

## Genotyping

The $Spo11^{Cas9-P2A-eGFP}$ knock-in allele was detected as in S7 Fig using primers and conditions in S1 Table. We detected inheritance of the $Tyr^{CopyCat}$ transgene by observing red fluorescence of mCherry (S5 Fig). We then genotyped all animals for the $Tyr^{ch}$ allele by PCR amplification of a 392-bp region containing the SNP followed by Sanger Sequencing (GeneWiz, CA, USA). Finally, we confirmed the $Tyr^{CopyCat}$ genotype in each $Tyr^{ch}$ animal using PCR primers that bind inside and outside of the transgene to amplify an 838-bp fragment only from transgenic animals (S6 Fig). PCR genotypes were performed according to conditions set in S1 Table.

The $Spo11^{Cas9-P2A-eGFP}$ knock-in allele was confirmed to be the correct sequence and integrated into the correct genomic location by 9 tiled PCR amplifications (S1 Fig) using primers

and conditions in S1 Table. The flanking amplicons, LHom and RHom, were each amplified using a primer that annealed inside the transgenic element (either the body of Cas9 or the body of eGFP) and a primer outside the homology arm in the genomic locus to confirm proper site-specific integration. Each of the 9 amplicons were then Sanger sequenced for verification (GeneWiz). All raw Sanger sequence traces are available at the associated Zenodo archive.

## Supporting information

**S1 Fig. PCR validation of *Spo11^{Cas9-P2A-eGFP}* integration. (A)** SnapGene schematic of P2A-Cas9-T2A-eGFP integrated into the 3′ end of the endogenous *Spo11* locus. LR, Left Homology Arm; RR, Right Homology Arm. Maroon arrows (1 to 7) represent aligned Sanger sequencing traces that tile the knock-in. Dark blue arrows (LHom/RHom) represent aligned Sanger sequencing traces that span the homology arm and part of the knock-in construct. Vertical dotted lines show that the primer binding sites are outside of the homology arm and inside of the synthetic construct. **(B)** PCR amplification of the left and right homology arm. All primer sequences, PCR conditions, and amplicon lengths are in S1 Table. The validated knock-in allele sequence is available at GenBank under accession number OK067273. Sanger sequencing files can be found at the associated Zenodo data repository (https://doi.org/10. 5281/zenodo.5510697) in folder "Spo11_Cas9_knockin_validation." The raw gel image in (B) can also be found at the Zenodo data repository labeled "S1 Raw Images."
(PDF)

**S2 Fig. Western blot of SPO11, CAS9, and eGFP in adult testis. (A–C)** Western blot of (A) SPO11, (B) CAS9, and (C) eGFP on 3 wild-type and 3 *Spo11^{Cas9-P2A-eGFP/+}* adult testes. **(A'–C')** Serially-stained membranes adding anti β-ACTIN to detect the loading control. PP, Precision Plus Ladder; MM, MagicMark Ladder. The raw gel images can be found at the associated Zenodo data repository (https://doi.org/10.5281/zenodo.5510697) in the file labeled "S1 Raw Images.pdf."
(PDF)

**S3 Fig. Hematoxylin and eosin staining of testes. (A)** *Spo11^{Cas9-P2A-eGFP/+}* and **(B)** *Spo11^{Cas9-P2A-eGFP/Cas9-P2A-eGFP}* seminiferous tubules; scale bar is 50 μm. Cells in seminiferous tubules in (B) likely do not complete meiosis, evidenced by absence of sperm. **(C)** Tiled image of *Spo11^{Cas9-P2A-eGFP/Cas9-P2A-eGFP}* testis; scale bar is 200 μm. All seminiferous tubules are deformed and devoid of sperm.
(PDF)

**S4 Fig. IF detection of eGFP in embryonic ovaries and mature testis. (A-C)** Wild-type embryonic ovary. All other panels in each row are *Spo11^{Cas9-P2A-eGFP/+}* littermates (e.g., A'–A‴). **(D)** Wild-type and **(D')** *Spo11^{Cas9-P2A-eGFP/+}* adult testis. Scale bar is 50 μm for all panels. Confocal fluorescence imaging settings are identical for all samples; eGFP expression is substantially higher in testes compared to any embryonic ovary time point. IF, immunofluorescence.
(PDF)

**S5 Fig. Detection of CopyCat transgene by red fluorescent mCherry in tail tips. (A)** *Tyr^{null}*/ *Tyr^{null}* tail tip. **(B, C)** *Tyr^{CopyCat}*/*Tyr^{null}*.tail tips. While clearly present or absent, mCherry expression varies in intensity between individuals of equivalent age.
(PDF)

**S6 Fig. PCR confirmation of *Tyr^{CopyCat}* in each offspring of an HDR event. (A)** Schematic of primer binding locations to detect presence of the *Tyr^{CopyCat}* transgene. **(B)** PCR confirmed presence of the *Tyr^{CopyCat}* transgene in each of the individuals marked as positive for a gene

conversion event, with families of offspring from males and females numbered as in the table in Fig 2D. Two gels were merged for ease of understanding. The raw gel images in (B) can also at the associated Zenodo data repository (https://doi.org/10.5281/zenodo.5510697) in the file labeled "S1 Raw Images."
(PDF)

**S7 Fig. Genotyping strategy for *Spo11*$^{Cas9-P2A-eGFP}$.** **(A)** Schematic of primer binding location for PCR genotyping. **(B)** Gel depicting PCR of Spo11 locus revealing the genotype of *Spo11*$^{+/+}$, *Spo11*$^{Cas9-P2A-eGFP/+}$, and *Spo11*$^{Cas9-P2A-eGFP/Cas9-P2A-eGFP}$. The raw gel image in (B) can be found at the associated Zenodo data repository (https://doi.org/10.5281/zenodo.5510697) in the file labeled "S1 Raw Images."
(PDF)

**S1 Table. Primer sequences and PCR conditions for each genotyping strategy.**
(PDF)

**S1 Raw Images. Full uncropped gel from S1B Fig.**
(PDF)

## Acknowledgments

We thank H. Cook-Andersen, M. Wilkinson, and R. Knight for discussions of strategy and meiotic drivers. V. Ruthig provided advice on germline immunofluorescence and staging of ovaries and testes. We thank D. Yelon for use of the Leica TCS SP8 multiphoton confocal microscope.

## Author Contributions

**Conceptualization:** Hannah A. Grunwald, Valentino M. Gantz, Stephen M. Hedrick, Ethan Bier, Kimberly L. Cooper.

**Data curation:** Alexander J. Weitzel.

**Formal analysis:** Alexander J. Weitzel.

**Funding acquisition:** Stephen M. Hedrick, Ethan Bier, Kimberly L. Cooper.

**Investigation:** Alexander J. Weitzel, Ceri Weber, Rimma Levina.

**Methodology:** Alexander J. Weitzel, Hannah A. Grunwald, Ceri Weber, Rimma Levina.

**Project administration:** Kimberly L. Cooper.

**Resources:** Valentino M. Gantz, Stephen M. Hedrick, Ethan Bier, Kimberly L. Cooper.

**Supervision:** Kimberly L. Cooper.

**Validation:** Alexander J. Weitzel.

**Visualization:** Alexander J. Weitzel, Ceri Weber, Rimma Levina.

**Writing – original draft:** Alexander J. Weitzel, Kimberly L. Cooper.

**Writing – review & editing:** Alexander J. Weitzel, Hannah A. Grunwald, Ceri Weber, Valentino M. Gantz, Stephen M. Hedrick, Ethan Bier, Kimberly L. Cooper.

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
