## [Editor Report · Decision Letter 0]

27 Apr 2021

Dear Kimberly, 

Thank you for submitting your manuscript entitled "Meiotic Cas9 expression mediates genotype conversion in the male and female mouse germline" for consideration as a Discovery Report by PLOS Biology. Please accept my apologies for the delay in getting back to you as we consulted with an academic editor about your submission. 

Your manuscript has now been evaluated by the PLOS Biology editorial staff as well as by an academic editor with relevant expertise and I am writing to let you know that we would like to send your submission out for external peer review.

Please re-submit your manuscript within two working days, i.e. by Apr 29 2021 11:59PM.

Kind regards,

Richard

Richard Hodge, PhD

Associate Editor, PLOS Biology

rhodge@plos.org

PLOS

---

## [Decision Letter · Decision Letter 1]

3 Jun 2021

Dear Dr Cooper,

Thank you very much for submitting your manuscript "Meiotic Cas9 expression mediates genotype conversion in the male and female mouse germline" for consideration as a Discovery Report at PLOS Biology. Please accept my apologies for the delay in getting back to you with our decision. Your manuscript has been evaluated by the PLOS Biology editors, an Academic Editor with relevant expertise, and by three independent reviewers. 

The reviews are attached below, as well as some specific comments from the Academic Editor in response to the reviewer reports. The Academic Editor comments are pasted below the reviews. You will see that the reviewers find your conclusions novel and interesting, but raise overlapping concerns that Spo11 function may have been reduced due to an intrinsic issue with the construct and ask for additional validation that the construct has been successfully integrated into the genome. In addition, the reviewers suggest additional experiments to support the conclusion that Cas9 expression levels govern germline conversion frequencies.

In light of the reviews, we will not be able to accept the current version of the manuscript, but we would welcome re-submission of a much-revised version that takes into account the reviewers' comments. We cannot make any decision about publication until we have seen the revised manuscript and your response to the reviewers' comments. Your revised manuscript is also likely to be sent for further evaluation by the reviewers. We expect to receive your revised manuscript within 3 months. 

In addition, having evaluated the manuscript and the reviewer reports once again, the editorial team feel that the manuscript would be a better fit as a Methods and Resources article instead of a Discovery Report. Upon re-submission, we ask that you please change the article type to a Methods and Resources article. 

**IMPORTANT - SUBMITTING YOUR REVISION**

*Re-submission Checklist*

*Published Peer Review*

*PLOS Data Policy*

*Blot and Gel Data Policy*

Sincerely,

Richard

Richard Hodge, PhD

Associate Editor, PLOS Biology

rhodge@plos.org

PLOS

REVIEWS:

Reviewer #1: In this manuscript, Weitzel and colleagues build upon their prior work in Cas9-mediated meiotic gene conversion to demonstrate a novel construct capable of inducing conversion in both the male and female germlines. 

The experiments are logically planned and carried through, and the breeding data presented are sufficient to show that at least some of the final generation offspring carry the CopyCat and chinchilla alleles on the same chromosome (Fig 2D, right hand column). Although the numbers are low, this demonstrates conclusively that a DSB was formed on the target chromosome, which then underwent some kind of HDR event with the donor (CopyCat) chromosome. 

My main technical query is in regard to characterising the Spo11-Cas9-GFP construct allele. Although they could not detect Cas9 by IF or WB, they could detect GFP by IF. Was GFP detectable by WB? This might give a clue as to the issues affecting protein expression. For example, if the P2A peptides are failing to cleave the expressed polyprotein, this could interfere with the folding and/or function of all three components and likely lead to substantial protein degradation. This would have the observed effects of a functional Spo11 knockout, low Cas9 expression, and GFP detectable only by IF and not by native fluorescence. A GFP WB would also provide better support for the claim that expression is higher in males - though I agree the supplementary figure is quite convincing in this regard.

Although not necessary for this paper, an obvious double check for all these results would be to use the a Spo11-Cre allele in conjunction with LSL-Cas9 to see if this gives higher expression of Cas9 in the desired cell types.

The remainder of the discussion is generally fine, but neglects the possibility that conserved alleles may have nevertheless been cut, but repaired via HDR from the sister chromatid. Although this is a rare possibility, it does occur for at least some breaks, perhaps particularly those occurring in late pachytene, diplotene or in secondary spermatocytes. Given the stronger expression of the male meiotic Cas9 in late pachytene/diplotene, this should be mentioned as a possibility.

Reviewer #2: Weitzel et al present a well written manuscript claiming:

1. meiotic expression of Cas9 mediates genotype conversion in the male as well as in the female germline. 

2. Cas9 expression levels may govern genotype conversion frequencies.

3. expression of Cas9 during meiosis I allows genotype conversion in the male and female germlines.

I find the ideas in the paper to be exciting and the potential applications (pest control gene drives, simplification of breeding strategies) to be high impact. However, I feel that the paper does not meet the scope of a 'Discovery Report' provided to me during the review process. The authors' argument is that they have a preliminary approach for germline genotype conversion and that this could be improved by increasing Cas9 expression or by refining timing of Cas9 expression. In my opinion, the authors do not do enough to justify future research in these directions and do not rule out alternative explanations. My recommendation is thus to revise the manuscript. My specific comments are below.

Major Concerns

1. I find the phenotype of the homozygous mouse to be concerning. To me, this implies that Spo11 in the fusion construct is damaged or repressed due to the extended transcript. The genotyping in Figure S1 does not rule out that this could have occurred. It would be nice to see better validation that the original construct inserted successfully into the genome (junction PCR and sequence validation, for example). It would also be nice to see some validation that Spo11 expression levels are relatively normal in the context of the -P2A-Cas9-P2A-eGFP fusion. This seems challenging in the context of tissue, but might be addressable by IF. My issue here is that the authors do not rule out the possibility that something intrinsic to their construct is the cause of lower than expected genotype conversion efficiency. 

2. The claim about Cas9 expression levels is interesting but needs to better supported to justify further investigation. One alternative explanation is some aspect of the original construct reduces Cas9 expression, as discussed above. Another alternative explanation would be silencing of the gRNA in the CopyCat cassette. I would want to see data indicating these alternative explanations were unlikely. Data (obtained using similar Cas9 constructs) suggesting that Cas9 levels higher than those measured here increased genotype conversion would also be conclusive. 

3. The claim that Meiosis I expression of Cas9 is important is compelling, but I find the definition of this timing in the manuscript to be somewhat circular. The authors cite expression patterns of Spo11 during meiosis, then use the expression of their GFP construct (driven by the Spo11 promoter) to define meiotic populations. At a minimum, the authors should use independent markers and demonstrate that their GFP expression patterns match expectations. Ideas in the discussion section to use promoters that "drive strong expression with less temporal specificity and to chemically control the timing of Cas9 activity" would also address this concern.

Comments

1. Cas9 antibodies are notoriously awful. Most people in the field use epitope tagged Cas9 for this reason. Some labs have had luck with Cell signaling 14697 and this clone from other vendors.

Reviewer #3: This manuscript describes additional "gene conversion" experiments from the group that previously published the first evidence for "gene drive" activity in mice (Grunwald et al). This research is highly topical - not least because of the potential for genetic biocontrol applications as well as generation of homozygous multi-allele disease models with greater efficiency. The study is very focussed - essentially comprising a single experiment that investigates germline HDR/EJ efficiency from a new Cas9-driver (Spo11-KI) allele. The elegant mating scheme used to detect gene conversion (which has the distinct advantage of a marked recipient chromosome) has been used previously (Grunwald et al). 

The manuscript is well written and the figures clear and informative. The following suggestions (many of which are minor) should be considered to improve the manuscript. 

1. P2 Line 34. The statemen of "72%" doesn't give a sense of the inter-individual variability seen previously in the Grunwald et al paper - suggested clarifying.

2. P3 line 4 "achieve timing" - could be expressed more clearly.

3. P3 Line 18 "SPO11" (ie. in CAPS) as the protein is being referred to.

4. P3 line 27 Although this issue is addressed in the discussion a caveat here is that DSB repair may be perfectly repaired and therefore not detected. Perhaps use "detectable DSB repair".

5. P5 line 31 - P7 line 4; The description of the mating scheme is slightly confusing because the Spo11-Cas9/+; Tyr ch/ch mice described in p7 line 2 cannot be generated by crossing Spo11-Cas9/+ with Tyr ch/ch mice (i.e. only Tyr ch/+ can be generated). Also, suggest including the male origin of the Spo11-Cas9 allele on Fig. 2. 

6. Despite the efforts not to reduce SPO11 activity, it appears that the P2A linker KI approach may have ablated/reduced SPO11 function. However, the only data provided to assess this is from a single HOM male animal (Fig S2). It would obviously be informative to include more data from HOM breeders if available. Also, corroborating evidence of reduced Spo11/SPO11 expression in HOM and HET Spo11-Cas9 mice would be very useful to include in the manuscript.

7. Given that SPO11 has a role in homologous recombination repair, if indeed SPO11 function has been reduced, it is possible that the Spo11-Cas9 HET mice used in the gene conversion test cross may have a reduced capacity for homologous recombination (aka HDR) - which is the mechanism used for gene conversion. This possibility should be acknowledged. 

8. Did you look for any evidence of "Cas9 carry-over" into the gametes (i.e. indel mutations in the Tyr-null allele)? Would these be indistinguishable from the "EJ" group? (Fig. 2D) 

9. The discussion is quite speculative. The model is based on estimation of expression levels and timing from different studies and techniques rather than direct comparison. That being said, I think it is a reasonable and likely correct interpretation of the available data and represents a valuable addition to our collective thoughts about of how gene conversion occurs and how it may be improved. As far as I am aware, careful comparison of Cas9 expression has not been performed even in Diptera (where gene conversion rates are generally much higher). 

10. Fig. S4 is only referred to in the Methods section. Could the authors please indicate if there was a perfect correlation between Cherry expression and a Tyr-Copycat allele-positive genotype.

Comments from the Academic Editor (AE):

REV #1: Although not necessary for this paper, an obvious double check for all these results would be to use the a Spo11-Cre allele in conjunction with LSL-Cas9 to see if this gives higher expression of Cas9 in the desired cell types.

AE: This might serve as good controls but I don’t think it is absolutely necessary.

REV #2: I find the ideas in the paper to be exciting and the potential applications (pest control gene drives, simplification of breeding strategies) to be high impact. However, I feel that the paper does not meet the scope of a 'Discovery Report' provided to me during the review process. The authors' argument is that they have a preliminary approach for germline genotype conversion and that this could be improved by increasing Cas9 expression or by refining timing of Cas9 expression. In my opinion, the authors do not do enough to justify future research in these directions and do not rule out alternative explanations. My recommendation is thus to revise the manuscript. My specific comments are below.

AE: I do not fully agree with this point. May we can ask for a better support with substantial addition of data? But for the purpose of ‘Discovery Report’, the current manuscript might suffice. Nevertheless, I still prefer to go for more conventional format if at all possible.

REV #2: 3. The claim that Meiosis I expression of Cas9 is important is compelling, but I find the definition of this timing in the manuscript to be somewhat circular. The authors cite expression patterns of Spo11 during meiosis, then use the expression of their GFP construct (driven by the Spo11 promoter) to define meiotic populations. At a minimum, the authors should use independent markers and demonstrate that their GFP expression patterns match expectations. Ideas in the discussion section to use promoters that "drive strong expression with less temporal specificity and to chemically control the timing of Cas9 activity" would also address this concern.

AE: The authors have made a knockin allele, so I am not concerned as much as this reviewer. I’d rather agree with Rev1 that the authors may use WB or IP/WB to detect the expression of Cas9 and eGFP in the knockin animals.

REV #2: 1. Cas9 antibodies are notoriously awful. Most people in the field use epitope tagged Cas9 for this reason. Some labs have had luck with Cell signaling 14697 and this clone from other vendors.

AE: This is a helpful comment. I’d definitely try to use a few antibodies to test the knockin allele with by WB or some other method than IF, which suffers a lot from autofluorescence from the mouse tissues.

REV #3: 6. Despite the efforts not to reduce SPO11 activity, it appears that the P2A linker KI approach may have ablated/reduced SPO11 function. However, the only data provided to assess this is from a single HOM male animal (Fig S2). It would obviously be informative to include more data from HOM breeders if available. Also, corroborating evidence of reduced Spo11/SPO11 expression in HOM and HET Spo11-Cas9 mice would be very useful to include in the manuscript.

AE: I’d also welcome any additional data but this is again optional. I wish that the authors discuss this point well in the discussion.

REV #3: 7. Given that SPO11 has a role in homologous recombination repair, if indeed SPO11 function has been reduced, it is possible that the Spo11-Cas9 HET mice used in the gene conversion test cross may have a reduced capacity for homologous recombination (aka HDR) - which is the mechanism used for gene conversion. This possibility should be acknowledged.

AE: Good point!

---

## [Decision Letter · Decision Letter 2]

10 Oct 2021

Dear Kim,

Thank you for submitting your revised Methods and Resources article entitled "Meiotic Cas9 expression mediates gene conversion in the male and female mouse germline" for publication in PLOS Biology. Please accept my apologies for the delay in getting back to you with our decision, especially in light of your upcoming application deadline. I have now obtained advice from the original reviewers and have discussed their comments with the Academic Editor.

The reviews are pasted below my signature. As you will see, the reviewers feel that the additional data in the revised manuscript has fully addressed their previous comments. Based on the reviews, we will probably accept this manuscript for publication, provided you address the following data and other policy-related requests that I have provided below. I am still very hopeful we can get your paper to editorial accept by end of this week once you have addressed these minor requests. We will do everything that we can to process your paper as quickly as possible once we receive the revision. 

(A) Please move the following ethics statement that you have provided in the manuscript to the Methods section. In addition, please provide the method of euthanasia used in the animal studies 

‘All animal research was conducted under UCSD Animal Welfare Assurance number D16-00020 19 and approved by Institutional Animal Care and Use Committee protocol number S14014.’

(B) You may be aware of the PLOS Data Policy, which requires that all data be made available without restriction: http://journals.plos.org/plosbiology/s/data-availability. For more information, please also see this editorial: http://dx.doi.org/10.1371/journal.pbio.1001797

- Supplementary files (e.g., excel). Please ensure that all data files are uploaded as 'Supporting Information' and are invariably referred to (in the manuscript, figure legends, and the Description field when uploading your files) using the following format verbatim: S1 Data, S2 Data, etc. Multiple panels of a single or even several figures can be included as multiple sheets in one excel file that is saved using exactly the following convention: S1_Data.xlsx (using an underscore).

- Deposition in a publicly available repository. Please also provide the accession code or a reviewer link so that we may view your data before publication.

Regardless of the method selected, please ensure that you provide the individual numerical values that underlie the summary data displayed in the following Figures, as they are essential for readers to assess your analysis and to reproduce it:

Figure 3A' and 3B'

(C) Please also ensure that each of the relevant figure legends in your manuscript include information on *WHERE THE UNDERLYING DATA CAN BE FOUND*, and ensure your supplemental data file/s has a legend

(D) Please ensure that your Data Statement in the submission system accurately describes where your data can be found and is in final format, as it will be published as written there. At this time, we ask that you please make sure the genotyping and sequence data deposited at the at the Zenodo (10.5281/zenodo.5510697) and Genbank (OK067273) databases are made publicly available. Please ensure that the accession number/DOIs provided are correct, as they do not appear to be functional at this stage. 

(E) We require the original, fully uncropped and minimally adjusted images supporting all blot and gel results reported in the following Figures:

S1, S2A-C, S6B, S7B

We will require these files before a manuscript can be accepted so please prepare and upload them now. Please carefully read our guidelines for how to prepare and upload this data: https://journals.plos.org/plosbiology/s/figures#loc-blot-and-gel-reporting-requirements.

We expect to receive your revised manuscript within two weeks. 

*Published Peer Review History*

*Early Version*

Sincerely,

Richard

Richard Hodge, PhD

Associate Editor, PLOS Biology

rhodge@plos.org

Reviewer Remarks:

Reviewer #1: Peter Ellis - note this reviewer has signed his review

The authors have answered all my queries. Nice work!

Reviewer #2: 

Thanks to the authors for their comprehensive response to reviewer concerns and for the positive tone of their response. I recommend publication. Look forward to seeing the print version.

Reviewer #3: Paul Quinton Thomas - - note this reviewer has signed his review

The authors have adequatly addressed all issues that I raised.

---

## [Editor Report · Decision Letter 3]

10 Nov 2021

Dear Kim,

On behalf of my colleagues and the Academic Editor, Bon-Kyoung Koo, I am pleased to say that we can in principle accept your Methods and Resources article "Meiotic Cas9 expression mediates gene conversion in the male and female mouse germline" for publication in PLOS Biology, provided you address any remaining formatting and reporting issues. These will be detailed in an email that will follow this letter and that you will usually receive within 2-3 business days, during which time no action is required from you. Please note that we will not be able to formally accept your manuscript and schedule it for publication until you have any requested changes.

PRESS

Sincerely, 

Richard

Richard Hodge, PhD

Associate Editor, PLOS Biology

rhodge@plos.org

PLOS
